# *IKZF1* Alterations and Therapeutic Targeting in B-Cell Acute Lymphoblastic Leukemia

**DOI:** 10.3390/biomedicines12010089

**Published:** 2024-01-01

**Authors:** Jonathan Paolino, Harrison K. Tsai, Marian H. Harris, Yana Pikman

**Affiliations:** 1Department of Pediatric Oncology, Dana-Farber Cancer Institute, Boston, MA 02215, USA; 2Division of Hematology/Oncology, Boston Children’s Hospital, Boston, MA 02115, USA; 3Department of Pathology, Boston Children’s Hospital, Boston, MA 02115, USAmarian.harris@childrens.harvard.edu (M.H.H.)

**Keywords:** *IKZF1*, IKAROS, B-cell acute lymphoblastic leukemia

## Abstract

*IKZF1* encodes the transcription factor IKAROS, a zinc finger DNA-binding protein with a key role in lymphoid lineage development. IKAROS plays a critical role in the development of lineage-restricted mature lymphocytes. Deletions within *IKZF1* in B-cell acute lymphoblastic leukemia (B-ALL) lead to a loss of normal IKAROS function, conferring leukemic stem cell properties, including self-renewal and subsequent uncontrolled growth. *IKZF1* deletions are associated with treatment resistance and inferior outcomes. Early identification of *IKZF1* deletions in B-ALL may inform the intensification of therapy and other potential treatment strategies to improve outcomes in this high-risk leukemia.

## 1. Introduction

Acute lymphoblastic leukemia (ALL) is the most common pediatric cancer, comprising approximately 25% of new cancer diagnoses in those under the age of 15 [1,2]. Childhood ALL represents a model treatment paradigm for pediatric cancer, integrating clinical and genomic features, as well as treatment response, to adjust therapy. This risk-adapted therapy has led to an overall survival (OS) in childhood B-ALL of greater than 90% [1,2]. Despite this success, acute leukemia remains the second leading cause of pediatric cancer-related death. Improved therapeutic approaches are needed for patients with treatment-refractory leukemia to both improve outcomes and decrease toxicity of highly intensive therapy. 

The initial genomic risk stratification of B-ALL is largely based on the detection of recurrent alterations associated with a more aggressive disease phenotype, informing the intensification of therapy or therapeutic targeting with a specific inhibitor. In B-cell ALL (B-ALL), these high-risk genomic features include hypodiploidy (<40 chromosomes), the intrachromosomal amplification of chromosome 21 (iAMP21), and translocations such as *TCF3::HLF*, *BCR::ABL1* (Philadelphia chromosome-positive (Ph+)), those involving *KMT2A*, and those associated with Philadelphia-like (Ph-like) ALL [3,4]. Deletions within the lymphoid transcription factor IKAROS family zinc finger 1 (*IKZF1),* either alone or in combination with other genomic features, have been independently associated with poorer outcomes in patients with B-ALL and have been recognized in some risk stratification algorithms [5,6,7]. Here, we review the function of IKAROS in lymphocyte development and its association with a treatment-refractory disease phenotype in B-ALL with the goal of delineating potential therapeutic treatment strategies to address this high-risk leukemia. 

## 2. IKAROS Structure and Function

The *IKZF1* gene, located on chromosome 7p12.2, comprises eight exons encoding the zinc finger DNA-binding protein IKAROS [8,9]. IKAROS is a lymphoid transcription factor belonging to a family of genes regulating lymphocyte differentiation, including HELIOS (*IKZF2*), AIOLOS (*IKZF3*), EOS (*IKZF4*), and PEGASUS (*IKZF5*) [9,10,11,12,13]. Members of the IKAROS family are structurally and functionally similar, maintaining highly conserved N-terminal zinc finger motifs, which are responsible for DNA binding and transcriptional regulation [9,14]. IKAROS is critical to lymphoid lineage differentiation and functional outcome, holding key roles in guiding the transition from a multipotent hematopoietic stem cell (HSC) to a B-cell precursor, and in regulating the B-cell proliferative response to antigenic stimuli [15,16,17]. 

Structurally, IKAROS is composed of six zinc finger regions. Four Krüppel-type zinc finger motifs, numbered F1 through F4 and encoded by exons 4 through 6, are located in the N-terminal region of the protein and are largely responsible for the DNA-binding properties of IKAROS [8,9]. F2 and F3 bind a pentameric core recognition sequence, GGGAA. The F1 and F4 motifs do not directly recognize this core DNA sequence but enhance the DNA binding of the IKAROS protein, and together at least three of the N-terminus zinc finger motifs are required for adequate DNA binding [9]. 

The C-terminus of IKAROS contains two zinc finger motifs, F5 and F6, encoded by exon 8 [18]. In contrast to the DNA-binding properties of the N-terminal zinc fingers, F5 and F6 have little DNA-binding potential and instead allow for dimerization between IKAROS proteins and enable other protein interactions [15,18]. These protein interactions are critical for IKAROS-mediated transcriptional regulation, allowing for high-affinity binding to a single core DNA recognition site and mediating interactions with distal regulatory sites [18]. 

Both the C- and N-terminal domains of IKAROS additionally contain conserved serine- and threonine-rich regions, through which post-translational phosphorylation events modulate DNA binding by IKAROS [19,20]. Phosphorylation in these regions changes throughout lymphocyte development, with increased phosphorylation resulting in reduced IKAROS DNA-binding activity [20]. Together, these events appear to guide DNA binding by IKAROS as lymphocytes progress through the cell cycle.

## 3. IKAROS Isoforms

IKAROS exists simultaneously in a number of different isoforms, formed through alternative splicing or by intragenic deletion [18,21]. In total, at least 16 isoforms have been described, which vary in the number of N-terminal zinc finger motifs, giving each isoform specific DNA-binding characteristics [18,22]. The presence and number of N-terminal domains confer differing affinities for recognizing sequences in promoters of target genes [15]. The differential binding affinities of the various IKAROS isoforms result in a range of transcriptional activities on target genes throughout lymphocyte development [15]. 

Three isoforms contain at least three N-terminal zinc finger motifs, the requisite number needed to bind DNA recognition sites [18]. These are IK1, which retains all four N-terminal zinc fingers (F1–F4), IK2 (F2–F4), and IK3 (F1–F3) (Figure 1A) [18,23]. IK1 and lK2 are able to enter the nucleus and are highly expressed throughout lymphocyte development [9,18]. Other isoforms, including IK4 (F2–F3), IK5 (F1), IK6 (no N-terminal zinc fingers), and IK7 (F4), lack the requisite number of zinc fingers to strongly bind DNA directly (Figure 1B) [18,23]. These isoforms, however, retain their C-terminal zinc finger motifs and influence lymphocyte development through dimerization with other IKAROS isoforms [15,18]. Dimer formation of non-DNA-binding isoforms with those that retain strong DNA-binding capacity results in a net reduction of IKAROS activity [18]. Together, these interactions modulate transcriptional regulation during lymphoid maturation [18]. 

## 4. Role of IKAROS Lymphoid Transcriptional Priming

IKAROS plays an important role in transcriptional priming of hematopoietic stem cells (HSCs), directing multipotent progenitors toward lymphoid lineage [16]. As a transcription factor, IKAROS exerts its vast regulatory influence on lymphoid differentiation through interactions with a wide array of genes critical to lymphocyte development. These include genes involved in chromatin remodeling and transcriptional regulation such as *SOX4*, *RUNX2*, *FOXP1*, and *HDAC9*, cell signaling receptors such as *IL7R*, *NOTCH1*, *FLT3,* and *CCR9*, and antigen receptor development genes such as *RAG1*, *DNTT*, *IGI*, and *IG6* [15,16,17]. Additionally, IKAROS influences lymphoid differentiation and proliferation via the regulation of downstream kinase signaling cascades [24].

In addition to the direct transcriptional regulation of genes critical to lymphocyte development, IKAROS is an important component of the nucleosome remodeling and deacetylase (NuRD) complex in lymphocytes [25]. The NuRD complex provides access to nucleosomes through Mi-2β, restricts chromatin through histone deacetylases, and guides chromatin remodeling in a lineage-specific manner [26,27]. IKAROS, in part, directs chromatin remodeling by tethering the NuRD complex to genes active in lymphoid differentiation [25,28]. IKAROS also exerts its influence on lymphoid lineage gene expression through NuRD-independent chromatin structural remodeling by guiding three-dimensional genome organization mediated through long-distance interactions with enhancers across multiple regulatory sites [29]. Through these interactions, IKAROS guides lymphoid lineage transcriptional priming and the regulation of HSC and progenitor-specific genes toward lymphoid lineage-restricted progenitors [16]. 

Illustrating the importance of the IKAROS protein in lymphoid differentiation, transgenic murine models with the *IKZF1* deletion of DNA-binding N-terminal zinc fingers F1–F3 demonstrated a complete absence of both gamma-delta and alpha-beta T-cells, as well as underdeveloped or absent thymic tissue [8]. Similarly, pro-, pre-, and mature B-cells were absent from the bone marrow and splenic tissue in this murine model [8]. There was no effect on non-lymphoid lineages, including myelocytes, monocytes, erythrocytes, megakaryocytes, and platelets, suggesting both the importance and specificity of IKAROS in lymphocyte development [8]. Together, these data suggest that the loss of IKAROS in HSCs deprives multipotent progenitors of the ability for lymphoid differentiation [8,17].

## 5. IKAROS Alterations in B-ALL: Deletions, Mutations, and Fusions

The most common *IKZF1* deletions are whole-gene deletions, intragenic deletions of exons 4–7 leading to the IK6 isoform, and intragenic deletions of exons 2–7 abolishing the usual ATG start codon in exon 2 [30,31]. Many other intragenic and partial deletions occur rarely but recurrently. The sequencing of the deletion breakpoints associated with IK6 has suggested that these deletions arise from aberrant RAG-mediated recombination [32]. IK6 and similar isoforms lack several or all N-terminal DNA-binding zinc finger motifs but retain their C-terminal domains, allowing for dimerization with other IKAROS family proteins. While unable to bind DNA or modulate transcription, this dominant negative isoform negatively regulates wild-type IKAROS within the cell, as well as other IKAROS family proteins like AIOLOS (Figure 1B) [18]. 

The loss of normal IKAROS function through mono-allelic deletion results in a dominant negative isoform and leads to the loss of the negative regulation of genes that limit self-renewal in lineage-restricted lymphoid progenitors. The deletion of *IKZF1* exon 5 in CD2-Cre transgenic murine lymphoid progenitor cells resulted in an isoform lacking sufficient N-terminal zinc fingers to bind DNA, mimicking the dominant negative isoforms reported in B-ALL. This led to arrested differentiation in a pre-B-cell stage characterized by the increased expression of genes involved in proliferation and self-renewal [24]. In this model, maturation arrest resulted in pre-B-cells with increased stromal adherence and proliferative potential. Indeed, the transplantation of these IKAROS-mutant pre-B-cells resulted in pre-B-ALL in NOD-SCID-Il2rg^−/−^ (NSG) mice, demonstrating leukemogenesis [24]. 

The frequency of genetic alterations in *IKZF1* is approximately 15% in Philadelphia chromosome (Ph)-negative pediatric B-ALL and up to 30% in National Cancer Institute (NCI) high-risk, Ph-negative B-ALL [6,33]. Among patients with IKAROS deletions, whole-gene deletions involving exons 1 through 8 are observed in 25–40%, deletions of exons 4–7 are observed in ~25%, and other partial deletions comprise the remainder [34]. 

In contrast to IKAROS deletions, mutations in *IKZF1* appear to be less common (Figure 1B). In the Children’s Oncology Group (COG) P9906 study, a study of children with high-risk, Ph-negative B-ALL, missense, frameshift, and nonsense mutations predicted to alter IKAROS function were detected in approximately 2% of patients [5,35]. Heterozygous *IKZF1* N159Y occurs in less than 1% of B-ALL and has been reported to disrupt normal IKAROS function by affecting the DNA-binding domain of IKAROS [36,37]. *IKZF1* N159Y, which is located in exon 5, is also recurrently associated with the partial tandem duplication of the mutated exon 5 [38,39]. These N159Y mutations result in the nuclear mislocalization of IKAROS and the induction of aberrant intercellular adhesion. This N159Y subtype results in an altered expression of genes involved in oncogenesis, chromatin remodeling, and signaling [40,41], with a subtype-defining, characteristic gene expression pattern [36].

Gene fusions with *IKZF1* have rarely been reported [42]. An analysis of 195 patients with B-ALL enrolled in the Nordic Society of Paediatric Haematology and Oncology (NOPHO) studies of ALL in 1992, 2000, and 2008 reported two in-frame gene fusions involving *IKZF1*, *IKZF1::SETD5*, and *IKZF1::NUTM1*, both retaining the DNA-binding domains of IKAROS [42,43]. *IKZF1* fusions and other alterations have been found to frequently co-exist with *ETV6* aberrations in the emerging category of B-ALL with an *ETV6::RUNX1*-like expression profile [42]. The *IKZF1::NUTM1* gene fusion retained the entire *NUTM1* coding region, which is notable as this fusion partner has been recurrently reported in B-ALL [44]. The NOPHO cohort has also reported two out-of-frame fusions, with the subsequent loss of the functional domains of IKAROS [42]. In contrast to deletions of *IKZF1*, the prognostic implication of gene fusions involving IKAROS is unknown.

Germline mutations in *IKZF1* have been associated with leukemia predisposition. In one study of 4963 patients with presumed sporadic ALL, 0.9% of cases were found to harbor 28 unique germline *IKZF1* mutations, including 25 missense, 2 nonsense, and 1 frameshift mutation [45]. While many of these mutations were located outside of known IKAROS DNA-binding or dimerization domains, in vitro modeling revealed that 79% demonstrated a functionally deleterious effect on drug sensitivity, cellular adhesion, or IKAROS localization [45]. Many of these mutations were hypothesized to interfere with non-DNA-binding roles of IKAROS, including participation in the SIN3 histone deacetylase, Mi-2/NuRD, and PRC2 complexes [45]. Additionally, transduction of Arf^−/−^ *BCR::ABL1* pre-B-cells with germline *IKZF1* variants resulted in a reduction in sensitivity to both dasatinib and dexamethasone in the majority of mutants, suggesting that germline *IKZF1* mutations may confer treatment resistance [45]. 

## 6. Prognosis of IKAROS Alterations in B-ALL

Partial deletions of *IKZF1* (*IKZF1^del^*) have been shown to be independently prognostic of relapse in B-ALL [5,6,7,46]. In the COG P9906 study, partial deletions or mutations of *IKZF1* were found in 30% of patients (*n* = 67), in whom they were associated with a 55% 5-year incidence of relapse when compared with 14% in those without an aberration of IKAROS [5,35]. In an independent cohort of 258 patients with B-ALL enrolled across multiple studies at St. Jude Children’s Research Hospital and including both high- and standard-risk clinical and genomic features, 19% were found to have an alteration involving *IKZF1*, which was associated with an increased risk of relapse at 5 years (46.3% vs. 22.5%) [5]. Gene-expression profiling showed higher expression of HSC signatures in leukemia with IKAROS deletions and a reduction in expression of B-cell lymphoid maturation signatures. 

The Dana-Farber Cancer Institute (DFCI) Consortium study 05-001 demonstrated *IKZF1* deletions in 16% of patients with newly diagnosed Ph-negative B-ALL [46]. *IKZF1* deletions were independently associated with a lower 5-year event-free survival (EFS) (63% versus 88%), a higher rate of induction failure (8% versus 1%), and a higher 5-year cumulative incidence of relapse (29% versus 8%) [46]. The negative prognostic implications of *IKZF1* deletions retained significance in multivariable models when adjusted for age, presenting white blood cell count, co-occurring genomic aberrations, and minimal residual disease at the end of induction [46].

The Dutch Childhood Oncology Group (DCOG) and German Cooperative ALL Group (COALL) studied outcomes for 1128 children with newly diagnosed ALL across five clinical trials (DCOG: ALL-8, ALL-9, and ALL-10 [47], and COALL: 06-97 [48] and 07-03 [49]). *IKZF1* deletions were identified in 16% of patients with Ph-negative, *KMT2A* wild-type B-ALL, which conferred a 5-year cumulative incidence of relapse of 34% compared with 13% in *IKZF1*-non-deleted patients [34]. 

*IKZF1* deletions are more frequent in leukemia occurring in people of Hispanic origin. In DFCI 05-001, *IKZF1* deletions were more likely to be present in patients of Hispanic ethnicity when compared with those of non-Hispanic ethnicity (30% vs. 13%) [46]. In a cohort of children with B-ALL at Children’s Hospital of Los Angeles, 29% of those with IKAROS deletions were of Hispanic ethnicity when compared with 15% in the non-Hispanic population [50]. Work is ongoing to understand the mechanism behind this ethnic disparity [51].

Further clarification of *IKZF1* deletion status resulted in the definition of the *IKZF1* plus (*IKZF1^plus^*) subtype as having a co-occurring deletion of *IKZF1* with the deletion of *CDKN2A*, *CDKN2B*, *PAX5*, or *PAR1* and in the absence of an ERG deletion [52,53]. An analysis of 1408 patients treated in the AIEOP-BFM ALL 2000 trial showed *IKZF1^plus^* in 6% of patients, in whom it was associated with a 5-year EFS of 53%, compared to 79% in patients with *IKZF1* deletion alone, and 87% in those without an *IKZF1* deletion [52]. High rates of relapse were observed in the *IKZF1^plus^* cohort with a cumulative 5-year relapse incidence of 44% versus 11% in those with *IKZF1* deletion alone [52]. In the European Organization for Research and Treatment of Cancer (EORTC) 58951, both *IKZF1^plu^*^s^ and *IKZF1^del^* were associated with a poor EFS, but outcomes were not statistically different between the groups [54,55,56]. In this cohort of 1200 children with Ph-negative B-ALL, *IKZF1* deletion was associated with a lower EFS of 75%, while *IKZF1^plus^* was associated with an EFS of 67%, versus 88% in *IKZF1*-non-deleted patients [54,56]. Despite this numerical difference, it did not rise to the level of significance [54]. It thus remains unclear whether *IKZF1* deletion alone or in combination with other genomic features should be used for prognostic determination and possible alteration of therapy. Methods for the evaluation of *IKZF1* status may influence detection of both *IKZF1* deletions and additional co-existing genetic alterations.

## 7. IKZF1 Deletions in the Context of Recurrent Mutations in B-ALL

*IKZF1* deletions co-occur with other B-ALL genomic aberrations, and the prognosis of these deletions in the context of other prognostic B-ALL alterations is variable. They are most common in Ph+ ALL (approximately 70% of patients) and are found in up to 68% of Ph-like ALL [4,57,58]. They can also be associated with *KMT2A* rearrangements (7%) and about 5% of *TCF3* rearrangements [34]. *IKZF1* deletions co-occur with traditionally standard-risk genomic subtypes as well, including in up to 15% of hyperdiploid and 3% of *ETV6::RUNX1* B-ALL [34]. 

In Ph+ ALL, IKAROS deletions occur in about 70% of patients [32,59,60,61]. IKAROS deletions in Ph+ ALL are associated with aberrant RAG-mediated recombination and are hypothesized to be a key event in leukemogenesis [32]. IKAROS deletions are an independent negative prognostic indicator in both adult and pediatric Ph+ ALL [60,61,62]. In a Ponte di Legno cohort of patients treated prior to the incorporation of the tyrosine kinase inhibitor imatinib into the therapeutic regimen for children with Ph+ ALL, *IKZF1* deletion conferred an inferior 4-year disease-free survival (DFS) of 30% versus 58% in those with wild-type *IKZF1* [61]. While DFS improved following the inclusion of imatinib for patients with Ph+ ALL, those with co-existing *IKZF1* deletion continued to have an inferior 4-year DFS of 56% versus 75% in those without an *IKZF1* deletion, confirming the negative prognostic impact of IKAROS deletions in children with Ph+ ALL [61]. 

As with Ph+ ALL, Ph-like ALL is also frequently associated with *IKZF1* alterations [3,4,5]. Ph-like ALL occurs in approximately 15% of pediatric B-ALL patients, in whom it is associated with a poor prognosis and is defined by a gene expression profile similar to Ph+ ALL in the absence of *BCR::ABL1* [4]. An array of mutations and gene fusions result in the Ph-like disease phenotype, comprising mutations involving the JAK/STAT pathway, including *CRLF2* rearrangements and *EPOR* fusions, ABL-class fusions (*ABL1*, *ABL2*, *CSF1R*, or *PDGFRA/PDGFRB*), activating RAS pathway mutations, or more rare kinase fusions involving *NTRK3*, *PTK2B*, and *DGKH,* among others [4,63,64]. Alterations in IKAROS are reported in up to 68% of Ph-like ALL [4,65]. In a study of 1589 patients enrolled under clinical trial protocols at St. Jude Children’s Research Hospital, COG, the Eastern Cooperative Oncology Group (ECOG), the Alliance for Clinical Trials in Oncology (Cancer and Leukemia Group B), and MD Anderson Cancer Center, both children and young adults with Ph-like ALL and co-existing *IKZF1* deletions demonstrated an inferior 5-year EFS of 48% in children and 19% in young adults, compared to 72% in children and 43% for young adults with Ph-like ALL without *IKZF1* deletion [4]. 

*IKZF1* deletions co-existing with favorable genetic B-ALL subtypes of *ETV6::RUNX1* or high hyperdiploidy (HeH) are somewhat less common. In a retrospective analysis of 939 patients with *ETV6::RUNX1* and 968 patients with HeH B-ALL across 16 clinical trials between 1991–2016, *IKZF1* deletions were detected in 3% of *ETV6::RUNX1* and 9% of HeH patients [66]. *IKZF1* deletions were associated with worse outcomes for these otherwise favorable prognostic subgroups. *ETV6::RUNX1* B-ALL with co-occurring *IKZF1* deletion was associated with a 5-year EFS of 79% (versus 92% in those without an *IKZF1* deletion), though measurable residual disease (MRD) stratification negated this adverse outcome for patients treated using MRD-guided protocols [66]. Patients with *IKZF1*-deleted HeH B-ALL had a 5-year EFS of 76% (versus 89% in *IKZF1*-non-deleted HeH B-ALL) even with MRD guided protocols, and *IKZF1^plus^* did not have an additional prognostic effect over *IKZF1^del^* alone [66]. 

It is notable that not all co-occurring genomic alterations associated with *IKZF1* deletion have negative prognostic implications. *DUX4* rearrangements (*DUX4_r_*) define a distinct subgroup of B-ALL, accounting for 3–7% of B-ALL cases [42,67,68,69]. They are associated with *IKZF1* deletions as well as the deletion of *ERG*. In this distinct subtype, rearrangement of *DUX4* to the immunoglobulin heavy chain (*IGH*) gene results in the overexpression of *DUX4* mediated by the *IGH* enhancer and is frequently associated with *ERG* deregulation or deletion [68]. For children enrolled in the EORTC- Children’s Leukemia Group (CLG) 58951 trial, *ERG*-deleted (*ERG^del^*) B-ALL was detected in 3.2% [70]. A co-occurring *IKZF1* deletion was present in 37.9% of *ERG^del^* cases versus 5.3% in the overall cohort. Among those with *IKZF1* deletion, the presence of a concomitant *ERG* deletion conferred a significantly higher 8-year EFS (85.7% versus 51.3% in those without DUX4_r_/ERG^del^), suggesting that the negative prognostic implications of *IKZF1* deletion did not extend to this genomic subtype [67].

## 8. Molecular Characterization of IKZF1 Alterations

There are many possible techniques available for the detection of *IKZF1* deletions, each with different strengths and weaknesses (Table 1). In general, the assay, or combination of assays, used to evaluate *IKZF1* deletion status in a clinical setting should be able to detect whole-gene deletions as well as exon-level (intragenic) deletions, with a clinically relevant turn around time. Multiplex ligation-dependent probe amplification (MLPA) has long been the gold standard for the detection of *IKZF1* deletions and is capable of detecting whole-gene deletions as well as single- and multi-exon deletions [71,72]. Fluorescence in situ hybridization (FISH) probes for *IKZF1* exist, but most probes are best suited to the detection of whole-gene rather than intragenic deletions. A recently described novel FISH assay has the potential to detect a subset of intragenic deletions but is not widely available [73]. DNA-based microarrays can be used to detect both whole-gene and intragenic deletions; the resolution is assay dependent and depends on probe coverage [74]. Polymerase chain reaction (PCR) assays have been designed to detect recurrent intragenic *IKZF1* deletions, and these highly sensitive assays have additional utility for evaluating MRD [75,76]. However, since these PCR assays target specific breakpoint clusters, they may miss rare deletions with unusual breakpoints [39].

Targeted DNA-sequencing panels that are validated for copy number detection and with adequate coverage over *IKZF1* can detect both whole-gene and intragenic deletions; the sensitivity and detectable deletion size varies by assay. DNA-sequencing assays have the advantage of also being able to detect sequence variants such as the subtype defining *IKZF1* N159Y alteration, as well as germline variants that may play a role in predisposition to B-ALL [45]. Although not yet available in most clinical laboratories, whole-genome sequencing has been shown to provide a comprehensive molecular characterization of pediatric B-ALL in the diagnostic setting, including the reliable identification of copy number alterations detected by MLPA as well as the identification of several events not called by MLPA, thus suggesting that MLPA cutoffs may be too stringent [38]. Moreover, whole-genome sequencing data allow for the identification of deletion breakpoints and may thus eventually enable effective estimates of the variant allele fractions of deletions, as sequencing continues to become cheaper and develop deeper coverage. 

Recently, multiple groups have also described using either whole-transcriptome RNA sequencing or targeted RNA sequencing (RNASeq) to detect *IKZF1* deletions [39,77,78,79,80,81]. Although these groups have used a variety of analytic approaches, in general *IKZF1* deletion detection from RNAseq data takes advantage of the novel transcripts generated from the deletion-containing allele, such as the junction between exon 3 and exon 8 in the transcript generated from an allele with the deletion of exons 4–7. This approach allows for higher sensitivity for low-level deletions than most DNA-based assays except for PCR. RNA sequencing has the advantage of the co-detection of clinically significant gene fusion events and gene expression levels (facilitating gene expression profiling), and can also detect expressed sequence alterations, including *IKZF1* N159Y. Moreover, total RNA sequencing without poly(A) enrichment retains pre-mRNA and intronic reads and has been shown to harbor the underlying genomic breakpoints of most *IKZF1* intragenic deletions [39]. On the other hand, RNA-based assays are not well-suited to detecting whole-gene deletions and may have slightly reduced sensitivity for out-of-frame deletions that result in nonsense-mediated decay. 

The performance of all these assays (except karyotype) is dependent on the specific probes and analytic pipelines used. In general, most of these assays are semi-quantitative and may report categorical rather than quantitative results; for larger *IKZF1* deletions, FISH and karyotype generally report the number of cells involved; however, these cytogenetic techniques may be subject to sampling and culturing conditions. Of note, most B-ALL diagnostic specimens have a high blast percentage, rendering the ability of a clinical assay to detect subclonal deletions non-essential; the clinical significance of subclonal *IKZF1* deletions at the time of diagnosis is still an area of active exploration [82].

## 9. Targeting *IKZF1*-Deleted B-ALL: Therapy Intensification and Novel Therapeutic Approaches

Intensification of conventional chemotherapy is a mainstay in the therapeutic targeting of high-risk ALL subtypes [1]. Based on the negative prognostic significance of *IKZF1*-deleted B-ALL, several trials have utilized this approach for patients with *IKZF1* deletions, including the DFCI ALL Consortium (NCT03020030), AIEOP-BFM (NCT03643276), DCOG [83], and the Malaysia-Singapore (MASPORE) leukemia study group [84] (Table 2). 

The Malaysia-Singapore ALL 2010 study prospectively intensified therapy for patients with identified *IKZF1* deletions and included imatinib for patients with Ph+ ALL [84]. Among 275 patients with B-ALL treated with this study protocol with sufficient diagnostic DNA for *IKZF1* characterization by MLPA, *IKZF1* deletions were detected in 18%. Those with high-risk disease, including patients with *IKZF1*-deleted B-ALL, received intensified therapy, including higher doses of methotrexate, three blocks of delayed intensification, and two additional chemotherapy cycles containing fludarabine, cytarabine, and daunorubicin [84,85]. Among patients with Ph-negative ALL, intensified therapy for those with *IKZF1* deletions was associated with an 11% cumulative incidence of relapse (CIR) and a 44% reduction in 5-year CIR compared to the previous trial (MS2003) [84]. Comparing the two trials, intensifying therapy for patients with *IKZF1* deletions, including the addition of imatinib for those with Ph+ ALL, improved the 5-year overall survival from 69.6% to 91.6% [84]. 

The DCOG ALL11 study demonstrated that prolonged maintenance therapy for patients with *IKZF1* deletions improved outcomes [83]. Children with *IKZF1*-deleted B-ALL received a third year of maintenance therapy consisting of methotrexate and intermittent 6-mercaptopurine in 3-week cycles [83]. Among those with *IKZF1* deletions and medium-risk MRD, prolonged maintenance therapy resulted in an EFS of 87.1% versus 72.3% in the preceding trial ALL10 and OS of 92.9% versus 83.0% in ALL10 [83]. 

Post-hoc analysis of the EORTC 58951 study revealed a significant improvement in DFS for children with *IKZF1*-deleted, Ph-negative B-ALL randomized to receive vincristine and steroid pulses during maintenance therapy [56]. Among 34 patients with *IKZF1* deletions included in the randomization, 15 were randomized to receive pulses during maintenance. The 8-year DFS for those who received vincristine and steroid pulses was the same for those with *IKZF1*-deleted and *IKZF1*-non-deleted ALL (93% versus 90%). Among those not receiving vincristine and steroid pulses during maintenance, the 8-year DFS was significantly worse for those with *IKZF1* deletion versus *IKZF1*-non-deleted (42.1% versus 89%) [56]. In contrast to these findings, however, a similar retrospective analysis for a cohort of patients enrolled in the ALL-BFM95 trial did not find improved outcomes for patients with *IKZF1* deletion receiving vincristine and steroid pulses during maintenance [86]. The benefit of this strategy remains unclear. 

IKAROS deletions co-occur with Ph+ and Ph-like targetable alterations. The success of tyrosine kinase inhibition in patients with Ph+ ALL extends to those with concomitant *IKZF1* deletions [87,88,89]. Ph-like ALL with ABL-class rearrangements (*ABL1*, *ABL2*, *CSF1R*, or *PDGFRB*) are sensitive to tyrosine kinase inhibition with imatinib and dasatinib, and these have been integrated in completed and ongoing trials, such as COG AALL1131 (NCT02883049), St. Jude Total Therapy XVII (NCT03117751), and COG AALL1631 (NCT03007147) [4,87,89]. COG AALL1521 (NCT02723994) is testing the role of JAK inhibition in Ph-like ALL with *EPOR* and *JAK2* rearrangements, or *CRLF2* rearrangements, which also co-occur with *IKZF1* deletion [90]. 

Novel approaches to addressing *IKZF1* deletions, partial or whole-gene, and their associated poor outcome in certain B-ALL subtypes, may center around overcoming the described chemotherapy resistance associated with these alterations. *IKZF1* loss is associated with the stem cell gene expression signature and may thus render cells more quiescent [91]. *IKZF1* loss has been associated with dexamethasone resistance in some studies, and this may be related to the degree of *IKZF1* loss [91,92,93,94]. In NALM6 cells, *IKZF1* knockout was associated with relative resistance to daunorubicin and asparaginase, while the Tanoue cell line with *IKZF1* knockout showed relative resistance to vincristine and asparaginase [91]. *IKZF1* regulates expression of *SAMHD1* in B-ALL, which may contribute to increased sensitivity to cytarabine in B-ALL with *IKZF1* deletions [91]. The expression of IK6 in a mouse model led to a decrease in dasatinib sensitivity in vivo, relevant to the co-occurrence of *IKZF1* deletions in Ph+ and Ph-like B-ALL [37]. How this chemotherapy resistance, especially with regards to levels of *IKZF1* expression, translates to in vivo sensitivity of ALL with *IKZF1* alterations is not fully known.

The direct targeting of IKZF1 to restore its function is not yet possible. The role of IKAROS in transcriptional priming of lymphoid progenitors is multifaceted, affecting gene expression across many cellular pathways, and it is possible to restore some downstream elements of normal IKAROS function in acute leukemia. Gene expression analysis of *IKZF1*-deleted cells showed potentially targetable changes, including in the JAK/STAT pathway [95]. Casein kinase 2 (CK2) inhibition restores transcriptional repression of the PI3K pathway, as well as genes important for cell cycle progression, in *IKZF1*-deleted acute leukemia [96,97]. Alterations in other signaling pathways, such as FLT3, will also need to be explored. *IKZF1* deletions have also been associated with altered apoptotic pathway control. IKAROS can transcriptionally repress *BCL2L1* (encodes BCL-XL) in B-ALL [98]. The inhibition of IKAROS by CK2 impairs the transcriptional repression of *BCL2L1*, associated with doxorubicin resistance. Targeting of the altered transcriptional program that results from *IKZF1* deletion may require the downstream inhibition of altered apoptotic pathways. 

Trials evaluating intensified therapy or the incorporation of targeted therapies for patients with IKAROS deletions will provide further insight into the role of these approaches. It remains to be seen whether the alteration of *IKZF1* alone or in combination with other genomic factors (like Ph-like or *IKZF1^plus^*) is the most relevant for prognosis.

## 10. Conclusions

IKAROS plays a key role in lymphoid lineage development by directing multipotent progenitors toward lineage-restricted mature lymphocytes. As a transcription factor, it exerts its regulatory influence on genes involved in chromatin remodeling, transcriptional regulation, cell signaling, and antigen receptor development. The loss of this regulatory influence limits the attenuation of HSC-specific properties in lymphoid lineage-restricted progenitors, including negative regulation of genes involved in self-renewal and growth. This persistent aberrant gene expression promotes the development of lineage-restricted lymphoid progenitors with a cancer stem cell phenotype. In B-ALL, mutations or deletions of *IKZF1* resulting in the loss of functional IKAROS are associated with inferior outcomes. The early identification of IKAROS alterations in B-ALL provides an opportunity to intensify therapy to potentially improve outcomes in this high-risk disease.

## Figures and Tables

**Figure 1 biomedicines-12-00089-f001:**
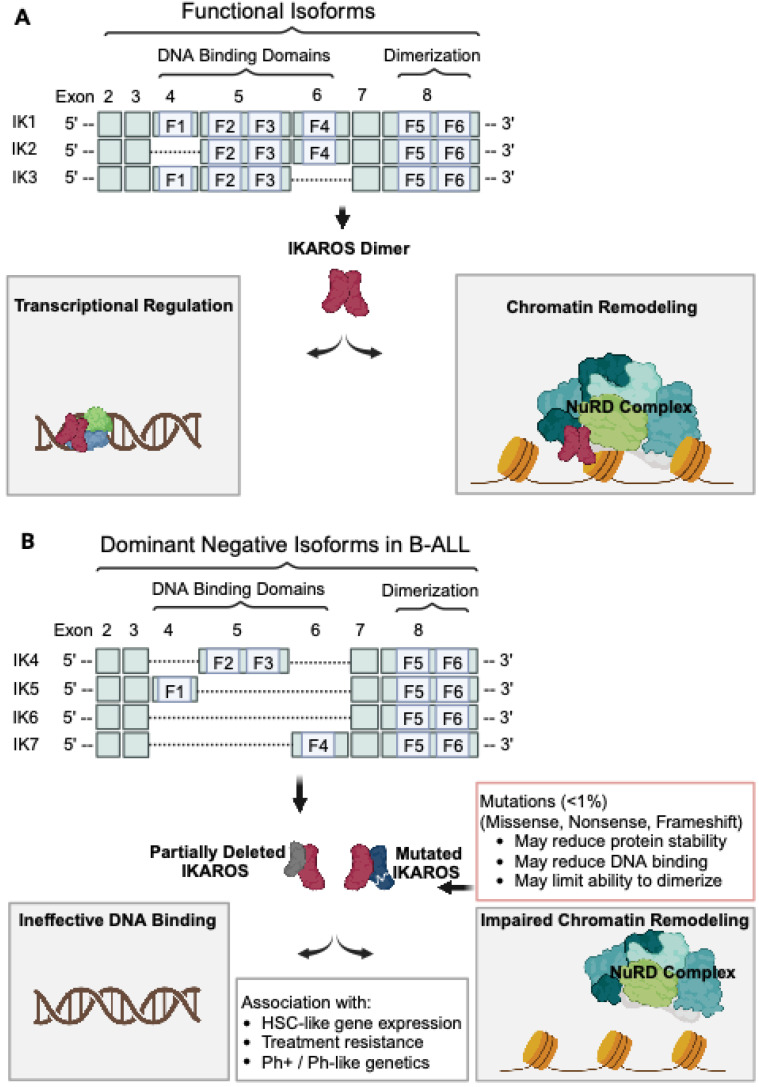
IKAROS deletions in B-ALL lead to ineffective transcriptional regulation and chromatin remodeling. (**A**) Functional IKAROS isoforms (IK1, IK2, and IK3; shown in red) retain at least three DNA-binding domains, which allow for effective DNA binding and transcriptional regulation. In B-cell progenitors, functional IKAROS dimers interact with other B-cell transcription factors, including EBF1, PAX5, E2A, and IRF4, and participate in core protein complexes such as the NuRD complex [15]. (**B**) Non-DNA-binding IKAROS isoforms (IK4, IK5, IK6, and IK7; shown in gray) retain the ability to dimerize, but lack the requisite number of binding domains to adequately bind DNA. Dimers formed with these isoforms demonstrate ineffective DNA binding and loss of transcriptional regulation. Missense, nonsense, or frameshift mutations (shown in blue) may reduce protein stability, limit DNA binding, alter nuclear localization, or impair dimerization with other IKAROS proteins, resulting in the loss of transcriptional regulation. Image created using BioRender.

**Table 1 biomedicines-12-00089-t001:** Comparison of assays for detection of IKZF1 aberrations. Multiplex ligation-dependent probe amplification (MLPA), polymerase chain reaction (PCR), and fluorescence in situ hybridization (FISH).

Technique	Substrate	Resolution	Detection of Subclonal Deletions or Cases with Low Tumor Burden	Comments
MLPA [71,72]	DNA	Exon level	Limited	Gold standard for detection of deletions
Microarray [74]	DNA	Exon level	Limited	
PCR [75,76]	DNA	Exon level	Yes	High sensitivity; potential for MRD detection
FISH [73]	DNA	Gene level	Down to ~5%	Inexpensive, fast turn around time
Karyotype	DNA	Chromosome band level (~5 Mb)	Down to ~10%	Unbiased evaluation of whole genome on individual cell basis; detection of rearrangements and other structural changes; requires viable cell culture
Targeted DNA sequencing [38]	DNA	Exon level	Limited	Detection of sequence variants
Targeted RNA sequencing or whole transcriptome sequencing [39,77,78,79,80,81]	RNA	Exon level	Yes	Detection of fusions; generates gene expression data; no detection of whole gene deletions

**Table 2 biomedicines-12-00089-t002:** Studies using *IKZF1* status to change chemotherapy regimen. Description of studies using *IKZF1* deletion for risk stratification and intensification of therapy. Measurable residual disease (MRD), event-free survival (EFS), and overall survival (OS).

Study	Genomic Feature	Intensified Therapy	Outcome
Malaysia-Singapore ALL2010 [84]	*IKZF1* deletion	Intensified therapy including two cycles of fludarabine, cytarabine, and daunorubicin, addition of imatinib for those with Ph+ ALL	OS of 91.6% vs. 69.6% in MS2003
DCOGALL11 [83]	*IKZF1* deletion	A third year of maintenance therapy(methotrexate and 6-mercaptopurine)	EFS of 87.1% vs. 72.3% and OS of 92.9% vs. 83.0% in ALL10
DFCI 16-001 (NCT03020030)	*IKZF1* deletion	Intensified to very high risk (VHR) therapy	Ongoing
AIEOP-BFM ALL2017 (NCT03643276)	*IKZF1^plus^* and MRD	Intensified to high risk (HR) therapy	Ongoing

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
