# Peer review of "IKZF1 Alterations and Therapeutic Targeting in B-Cell Acute Lymphoblastic Leukemia"

_biomedicines, 2024, doi:10.3390/biomedicines12010089_

Round 1
Reviewer 1 Report
Comments and Suggestions for Authors
The manuscript "Targeting IKZF1 alterations in B-cell acute lymphoblastic leukemia", written by Paolino J, Tsai HK, Harris MH and Pikman Y. is a review aiming to present the outcomes of different types of B-cell acute lymphoblastic leukemia in relation to the mutations of Ikaros and other mutations present, as well as therapies applied. The main massage is that the early identification of Ikaros deletions could improve the outcome by adjusting the therapy.
In the Introduction there is a short presentation of B-ALL, followed by description of Ikaros structure and functions, its isoforms and roles in lymphocyte development. Also, analysis of patients' outcomes in the cases of Ikaros alterations in B-ALL is done. Different methods for detection of Ikaros mutations are presented, as well as studies of therapy intensification and development of chemotherapy resistance associated with Ikaros alterations are described.
The topic is relevant to the field. Although there are numerous studies on B-ALL and Ikaros, the manuscript is an overview of trials and studies, with the numerical data supporting the main idea.
Conclusions are consistent and references appropriate.
Tables and figures are appropriate.
Other comments
There is a recent article of Hu et al. (Cell, 2023) describing the role of Ikaros in 3D genome organization, which could be mentioned when describing Ikaros roles.
Author Response
Reviewer #1:
The manuscript "Targeting IKZF1 alterations in B-cell acute lymphoblastic leukemia", written by Paolino J, Tsai HK, Harris MH and Pikman Y. is a review aiming to present the outcomes of different types of B-cell acute lymphoblastic leukemia in relation to the mutations of Ikaros and other mutations present, as well as therapies applied. The main message is that the early identification of Ikaros deletions could improve the outcome by adjusting the therapy.
In the Introduction there is a short presentation of B-ALL, followed by description of Ikaros structure and functions, its isoforms and roles in lymphocyte development. Also, analysis of patients' outcomes in the cases of Ikaros alterations in B-ALL is done. Different methods for detection of Ikaros mutations are presented, as well as studies of therapy intensification and development of chemotherapy resistance associated with Ikaros alterations are described.
The topic is relevant to the field. Although there are numerous studies on B-ALL and Ikaros, the manuscript is an overview of trials and studies, with the numerical data supporting the main idea.
Conclusions are consistent and references appropriate.
Tables and figures are appropriate.
Thank you for the thorough review and helpful feedback.
Other comments
There is a recent article of Hu et al. (Cell, 2023) describing the role of Ikaros in 3D genome organization, which could be mentioned when describing Ikaros roles.
Thank you for bringing our attention to this recent publication. We have incorporated this reference into the manuscript discussion beginning at line 130 and it is now included as reference number 29.
Reviewer 2 Report
Comments and Suggestions for Authors
In this review article, the authors reviewed the function of IKAROS in lymphocyte development and its association with a treatment-refractory disease phenotype in B-ALL with the goal of delineating potential therapeutic treatment strategies to address this high-risk leukemia.
Comments
This is an interesting review article. The manuscript is well-writing. The reviewer has only some minor concerns as follows:
1. Regarding the IKAROS structure description section, attaching a protein structure diagram can enhance readers' understanding and interest.
2. In Figure 1B, the text descriptions (Partially Deleted IKAROS and Mutated IKAROS) can be separated from the images for IKAROS protein to make it easier to read.
3. In Tables 1 and 2, the relevant references can be listed in the table to facilitate readers' search for relevant information.
4. In the Title, it does not require a period.
Author Response
Reviewer #2:
In this review article, the authors reviewed the function of IKAROS in lymphocyte development and its association with a treatment-refractory disease phenotype in B-ALL with the goal of delineating potential therapeutic treatment strategies to address this high-risk leukemia.
Comments
This is an interesting review article. The manuscript is well-writing. The reviewer has only some minor concerns as follow
- Regarding the IKAROS structure description section, attaching a protein structure diagram can enhance readers' understanding and interest.
Thank you for this interesting suggestion. The protein structure of IKAROS has been published however we believe that the gene level diagram of the protein motifs is more clear to understand and include it in Figure 1.
- In Figure 1B, the text descriptions (Partially Deleted IKAROS and Mutated IKAROS) can be separated from the images for IKAROS protein to make it easier to read.
We have edited the text description within Figure 1B.
- In Tables 1 and 2, the relevant references can be listed in the table to facilitate readers' search for relevant information.
We have added the references to the tables.
- In the Title, it does not require a period.
We have updated the title of the manuscript based on the suggestion of Reviewer #4. The title no longer has a period.
Reviewer 3 Report
Comments and Suggestions for Authors
This is a very well written Review regarding the IKAROS deletion and its' significance in B'ALL and i have the following concerns for the authors.
1. Is it possible to target the deranged IKAROS deletion in B-ALL in order to restore its' normal function? Are there any clinical trials on this or experiments in the lab? PLease add a paragraph to the discussion.
2. If not, in the previous question, why is it not possible and what are the main problems known by science until today for targeting IKAROS? Please add a novel paragraph in the discussion.
3. Are there any other reported in the literature mutations/molecular lesions or cytogenetic aberrations co-existing with IKAROS defects? Please add a novel paragraph in the discussion.
4. Except for B-ALL, are there other B-cell lympoproliferative disorders harboring the IKAROS molecular defects? Please add a novel paragraph to the discussion.
Overall, the authros have done a great work.
Author Response
Reviewer #3
This is a very well written Review regarding the IKAROS deletion and its' significance in B-ALL and I have the following concerns for the authors.
- Is it possible to target the deranged IKAROS deletion in B-ALL in order to restore its' normal function? Are there any clinical trials on this or experiments in the lab? Please add a paragraph to the discussion.
Thank you for this interesting suggestion. There are no targeted therapies that have been shown to restore normal IKAROS function in IKAROS deleted ALL. There are, however, targeted therapies available that function to restore some of the downstream effects of IKAROS in lymphocyte transcriptional priming. The most well described is that of casein kinase 2 (CK2). There are some preclinical data which suggest that CK2 inhibition may restore transcriptional repression of the PI3K pathway as well as genes important for cell cycle progression. This has been added to the discussion section beginning at line 446.
- If not, in the previous question, why is it not possible and what are the main problems known by science until today for targeting IKAROS? Please add a novel paragraph in the discussion.
Restoring normal protein function for gene deletions is challenging. There is no direct way to restore IKAROS function in IKZF1 deleted ALL, as discussed in comment 1. There are, however, ways to restore some of the downstream effects of IKAROS. This has been added as discussed in comment 1.
- Are there any other reported in the literature mutations/molecular lesions or cytogenetic aberrations co-existing with IKAROS defects? Please add a novel paragraph in the discussion.
Yes, there are other mutations, molecular lesions and cytogenetic alterations that co-exist with IKZF1 deletions. This has been outlined in the section titled “IKZF1 Deletions in the Context of Recurrent Mutations in B-ALL”.
- Except for B-ALL, are there other B-cell lymphoproliferative disorders harboring the IKAROS molecular defects? Please add a novel paragraph to the discussion.
While the role of IKAROS alterations in B-ALL and acute lymphoblastic lymphoma are well described, deletions of IKAROS appear to have little impact in mature B cell lymphomas and other B cell lymphoproliferative disorders. There are few reports in the literature describing IKAROS alterations on other B cell lymphoproliferative disorders. A case of Burkitt lymphoma associated with a germline mutation involving the dimerization domain of IKAROS has been reported (Kuehn HS, et al. Germline IKAROS dimerization haploinsufficiency causes hematologic cytopenias and malignancies. Blood. 2021). IKAROS gene expression has been reported to be the same across Hodgkin, anaplastic large cell, diffuse large cell and follicular center cell lymphomas compared to control normal lymphoid tissue (M. Antica, et al. Aberrant Ikaros, Aiolos, and Helios expression in Hodgkin and non-Hodgkin lymphoma. Blood 2008). Diffuse large cell lymphomas characterized by t(3;7)(q27;p12) results in the fusion between IKZF1 and BCL6 (Hosokawa Y et al Blood 2000). We suspect that this difference in the impact of IKAROS in other lymphoid diseases relates to the immature nature of lymphoblastic leukemia as compared to other more mature B-cell lymphoproliferative disorders.
Overall, the authors have done great work.
Reviewer 4 Report
Comments and Suggestions for Authors
The authors extensively reviewed the normal structure and function of IKZF1, its alterations in BALL, and influences of the alterations on therapeutic responses and prognosis of childhood B-ALL, and also therapeutic strategies based on the gene alterations. Although this review is informative and educational for hematologists, the Title, Abstract/Conclusions should be improved.
Major comments:
1. The title should be changed such as “IKZF1 alterations in B-cell acute lymphoblastic leukemia and therapeutic strategies including molecular targeting” because the authors mainly reviewed intensified chemotherapy protocols based on the kinds of gene alteration but not IKZF1-targeted agents, although they mentioned TKIs and JAK-inhibitors to some extent.
2. Abstract/Conclusions: The third sentence is inadequate, because it makes an impression that deletions within IKZF1confer normal stem cell properties to B-ALL cells. The second half of the sentence should be changed such as “conferring leukemic stem cell properties including self-renewal and subsequent uncontrolled growth to B-ALL cells. Also please consider whether or not the third sentence is appropriate as important component in the Abstract/Conclusions of this review.
Minor comments:
1. line 325: FISH should be written in full-term, then abbreviate.
2. Table 1: MLPA, PCR, FISH should be described in the same way as the legend for Table 1.
3. lines 352 to 356: Please make sentences without parentheses.
4. line 106: NuRD complex (15).→NuRD complex [15].
5. line 107: dimerize however lack→dimerize, however lackv
Comments on the Quality of English LanguagePlese refer to minor comments.
Author Response
Reviewer #4
The authors extensively reviewed the normal structure and function of IKZF1, its alterations in BALL, and influences of the alterations on therapeutic responses and prognosis of childhood B-ALL, and also therapeutic strategies based on the gene alterations. Although this review is informative and educational for hematologists, the Title, Abstract/Conclusions should be improved.
Major comments:
- The title should be changed such as “IKZF1alterations in B-cell acute lymphoblastic leukemia and therapeutic strategies including molecular targeting” because the authors mainly reviewed intensified chemotherapy protocols based on the kinds of gene alteration but not IKZF1-targeted agents, although they mentioned TKIs and JAK-inhibitors to some extent.
Thank you very much for these comments. We agree that there are no targeted therapies that directly restore IKAROS function in B-ALL. IKAROS degraders are under study in myeloid leukemias however in B-ALL it is the loss of normal IKAROS function that drives the phenotype. For this reason the main approaches to targeting IKAROS deletions in B-ALL include intensification of therapy, targeting of co-occurring mutations and targeting of increased signaling of other pathways that result from IKZF1 deletion. We have updated the title according to your recommendations and to better reflect the article.
- Abstract/Conclusions: The third sentence is inadequate, because it makes an impression that deletions within IKZF1confer normal stem cell properties to B-ALL cells. The second half of the sentence should be changed such as “conferring leukemic stem cell properties including self-renewal and subsequent uncontrolled growth to B-ALL cells. Also please consider whether or not the third sentence is appropriate as important component in the Abstract/Conclusions of this review.
We appreciate this comment and the abstract has been updated to include the suggested change. We do feel this is important as it highlights the stem cell like properties that are thought to confer a phenotype that is more resistant to therapy.
Minor comments:
- line 325: FISH should be written in full-term, then abbreviate.
We have written out FISH completely in the revised version.
- Table 1: MLPA, PCR, FISH should be described in the same way as the legend for Table 1.
We have added the description in the revised version.
- lines 352 to 356: Please make sentences without parentheses.
This has been corrected in the revised version.
- line 106: NuRD complex (15).→NuRD complex [15]
This has been corrected in the revised version.
- line 107: dimerize however lack→dimerize, however lack
This has been corrected in the revised version.
Round 2
Reviewer 3 Report
Comments and Suggestions for Authors
I have no further concerns.
Reviewer 4 Report
Comments and Suggestions for Authors
Thank you for the revision of the fomer manuscript.
The present manuscript has been resonably revised.